# Repair of Rat Calvarial Critical-Sized Defects Using Heparin-Conjugated Fibrin Hydrogel Containing BMP-2 and Adipose-Derived Pericytes

**DOI:** 10.3390/bioengineering11050437

**Published:** 2024-04-29

**Authors:** Gulshakhar Kudaibergen, Sholpan Mukhlis, Ainur Mukhambetova, Assel Issabekova, Aliya Sekenova, Madina Sarsenova, Abay Temirzhan, Murat Baidarbekov, Baurzhan Umbayev, Vyacheslav Ogay

**Affiliations:** 1Stem Cell Laboratory, National Center for Biotechnology, Astana 010000, Kazakhstan; kudaibergen@biocenter.kz (G.K.); muhlis@biocenter.kz (S.M.); mukhambetova@biocenter.kz (A.M.); issabekova@biocenter.kz (A.I.); a.sekenova@biocenter.kz (A.S.); sarsenova@biocenter.kz (M.S.); 2National Scientific Center of Traumatology and Orthopedics Named after Academician N.D. Batpenov, Astana 010000, Kazakhstan; abaytemirzhan@gmail.com (A.T.); b.m.u.80@mail.ru (M.B.); 3Laboratory of Bioengineering and Regenerative Medicine, Center for Life Sciences, National Laboratory Astana, Nazarbayev University, Astana 010000, Kazakhstan; bauyrzhan.umbayev@nu.edu.kz

**Keywords:** fibrin hydrogel, heparin, pericytes, regeneration, bone morphogenic protein, critical-sized bone defect

## Abstract

The repair of critical-sized calvarial defects is a challenging problem for orthopedic surgery. One of the promising strategies of bone bioengineering to enhance the efficacy of large bone defect regeneration is the combined delivery of stem cells with osteoinductive factors within polymer carriers. The purpose of the research was to study the regenerative effects of heparin-conjugated fibrin (HCF) hydrogel containing bone morphogenetic protein 2 (BMP-2) and adipose-derived pericytes (ADPs) in a rat critical-sized calvarial defect model. In vitro analysis revealed that the HCF hydrogel was able to control the BMP-2 release and induce alkaline phosphatase (ALP) activity in neonatal rat osteoblasts. In addition, it was found that eluted BMP-2 significantly induced the osteogenic differentiation of ADPs. It was characterized by the increased ALP activity, osteocalcin expression and calcium deposits in ADPs. In vivo studies have shown that both HCF hydrogel with BMP-2 and HCF hydrogel with pericytes are able to significantly increase the regeneration of critical-sized calvarial defects in comparison with the control group. Nevertheless, the greatest regenerative effect was found after the co-delivery of ADPs and BMP-2 into a critical-sized calvarial defect. Thus, our findings suggest that the combined delivery of ADPs and BMP-2 in HCF hydrogel holds promise to be applied as an alternative biopolymer for the critical-sized bone defect restoration.

## 1. Introduction

The restoration of large bone defects continues to pose significant challenges and remains an unresolved issue in orthopedic surgery. Critical-sized bone defects after bone cancer resection or severe trauma do not repair spontaneously and require a significant amount of bone grafting [1,2]. At present, for critical-sized bone defects treatment, various osteoplasty techniques using bone substitutes, autologous and allogeneic bone grafts are applied [3,4,5]. The use of bone autografting is considered as the “gold standard” in the treatment of bone defects. Nevertheless, autografts have some drawbacks in clinical practice: (1) donor bone site harvesting is a painful procedure and requires a significant amount of time for recovery; (2) there is a lack of the required amount of donor bone tissue. In situations where autografting is challenging, bone allografting emerges as the most common alternative. This method involves using cadaveric bones, eliminating the need for additional surgical procedures. However, there are significant disadvantages associated with this approach. Among the disadvantages are decreased bone strength due to sterilization, the potential for allograft rejection, and the risk of contracting infections. The application of bone graft substitutes does not always result in the successful restoration of large bone defects. In this regard, the great hopes in large bone defect repair are reasonably associated with the application of biomaterials for the effective repair of damaged bone tissues using osteoprogenitor cells, bioactive molecules and scaffolds.

Among all biomaterials, hydrogels have attracted the most widespread interest, particularly in their application as scaffolds in bone tissue engineering. This is primarily due to their structural resemblance to the extracellular matrix (ECM) and their porous nature, which renders them highly suitable for delivering cells [6]. Hydrogels are intricate three-dimensional (3D) networks that are cross-linked and formed by hydrophilic homopolymers, copolymers, or macromers. They exhibit swelling properties when placed in solutions, thereby creating a conducive microenvironment akin to extracellular matrices (ECMs). This environment facilitates various processes such as the migration, adhesion, proliferation, and differentiation of stem cells into bone-forming cells. Additionally, hydrogels serve as effective mediums for delivering essential nutrients and bioactive molecules [7,8,9]. In recent years, injectable hydrogels have garnered increasing attention from both scientists and clinicians as a promising biomaterial for tissue engineering. This is primarily due to their capacity to be implanted in a minimally invasive manner and their ability to conform to any desired shapes, corresponding to irregular defects [10].

Effective biomaterials and suitable manufacturing techniques are pivotal in the advancement of injectable hydrogels designed to serve as scaffolds for bone tissue engineering. In the past decade, there has been significant development in the creation of various types of injectable hydrogels. These biomaterials comprise both natural and synthetic polymers, including alginate, fibrin, collagen, chitosan, poly-L-lactic acid, hyaluronic acid and polyethylene glycol [11]. Both physical and chemical synthesis techniques have been utilized to create injectable hydrogels [12,13]. While injectable hydrogels for bone tissue repair have been developed using various methods over the years, there are still unresolved issues that need attention in the creation of functionalized injectable hydrogels for effective bone repair and regeneration. In the search for tissue-engineered techniques for manufacturing appropriate biomaterials, several challenges arise: selecting the optimal cell source, identifying growth factors that stimulate regeneration, and determining the ideal biodegradable and biocompatible carrier.

A promising cell source for tissue engineering is adipose tissue, which, when compared to other sources, harbors a significant number of not only mesenchymal stem cells (MSCs) but also perivascular cells, such as pericytes [14]. Pericytes are a unique population of perivascular cells found not only in adipose tissue but also in various other organs and tissues throughout the body (brain, skeletal muscle, bone marrow, placenta) and express both perivascular cell markers and mesenchymal stem cell markers [15]. Pericytes have multilinear differentiation capacity, particularly in the osteogenic direction. Additionally, pericytes produce angiogenic factors and actively participate in angiogenesis, suggesting that they may play a crucial role in the vascularization of bone tissue following injury [16]. Indeed, in vivo studies showed that CD146+ perivascular cells transplantation using an osteoinductive demineralized bone matrix into animal muscle tissue significantly increased graft ectopic ossification compared with samples containing SVF cells [17]. In addition, it was revealed that increased ossification in implants containing perivascular cells was accompanied by an increase in vascularization due to the production of vascular endothelial cell growth factor (VEGF) by perivascular cells [18]. In another study, adipose tissue-derived pericytes were implanted into a polymer scaffold coated with hydroxyapatite to obtain an additional osteoinductive effect. The results showed that a polymeric scaffold with pericytes was more effective in accelerating regeneration in a rat skull with bone defects as compared to a polymeric scaffold with stromal vascular fraction cells [19]. Tawonsawatruk and colleagues in their study using the rat tibia atrophic nonunion model showed that the transplantation of CD146+ pericytes from human adipose tissue significantly increased bone callus and mineralization, which led to an accelerated healing of rat tibia fracture [19]. Thus, all these findings indicate the powerful regenerative potential of pericytes in bone tissue regeneration.

Another important property of biomaterial is osteoinduction. Currently, proteins from the bone morphogenetic proteins (BMP) family are used to create osteoinductive biomaterials. BMPs are among the key factors in damaged bone tissue repair [20]. These proteins exhibit significant osteoinductive effects and possess the ability to enhance the formation of new bone tissue by promoting the differentiation of progenitor cells into osteoblasts [21]. Several recombinant BMPs have been approved by the FDA and are currently being used in medical practice to treat bone defects. However, despite the high efficacy of BMPs, there are still some concerns regarding their clinical application, which are regarding the short lifespan of BMPs. Proteins administered into the injury site lose their activity relatively quickly. Thus, in order to achieve therapeutic effect, significant concentrations of BMPs are applied in medical practice [22]. High doses of BMPs (1–2 mg/mL) can diffuse from the injury site and induce side effects, such as abnormal bone overgrowth and inflammation [23]. To avoid these problems, several drug delivery systems of BMP-2 have been proposed. For example, Mumcuoglu et al. created an injectable hydrogel using collagen microspheres and alginate for delivering BMP-2 to bone defects [24]. They found that the application of injectable hydrogel with 50 μg/mL BMP-2 was most effective in the regeneration of rat cranial defects. Other researchers, considering the heparin-binding ability of BMPs, incorporated heparin into scaffolds such as chitosan, poly-L-lactic acid, and a demineralized bone matrix [25]. It has been shown that heparin addition to the scaffolds stabilized recombinant BMP-2 and increased osteogenesis in vivo. Similar research was carried out by Yang et al., who developed an injectable fibrin hydrogel for the sustained delivery of BMP-2. They achieved this by covalently conjugating heparin to fibrinogen and demonstrated its efficacy in regenerating bone defects [26].

The heparin-conjugated fibrin (HCF) hydrogel combines the characteristics of fibrin hydrogel as a biocompatible scaffold with heparin, which is an anticoagulant. This combination has several benefits, especially in the fields of regenerative medicine and tissue engineering. First, fibrin hydrogel serves as a three-dimensional scaffold that imitates the extracellular matrix, enhancing cell adhesion and migration. Heparin may improve these characteristics by interacting with cell surface receptors to promote cell adhesion and migration [27]. Second, the anticoagulant and growth factor binding characteristics of heparin make HCF hydrogel suitable for drug delivery. This property is beneficial in tissue engineering, since it enables the continuous release of bioactive molecules that stimulate cell growth and tissue regeneration. Third, heparin has been found to have a proangiogenic impact by stimulating the development of new blood vessels [28]. In bone tissue engineering, this may help ensure adequate vascularization of the implanted construct, aiding in the transportation of nutrients and oxygen to the cells.

Therefore, based on these literature data, we supposed that the combinatory delivery of BMP-2 and adipose-derived pericytes (ADPs) via HCF hydrogel could enhance the healing and regenerative effects in critical-sized calvarial defects due to increased osteogenic differentiation of ADPs. Thus, the purpose of this research was to study the regenerative effects of HCF hydrogel containing BMP-2 and ADPs to repair critical-sized calvarial defects in the rats.

## 2. Materials and Methods

### 2.1. Animals

Male Wistar rats (10–12 weeks old, 260–320 g) were purchased from the Laboratory Animal Company “Pushchino” (Pushchino, Russia). The rats were housed in vivarium settings at 23 °C and 60% relative humidity. The animals were provided with ad libitum access to food and water. All procedures involving animals were conducted in strict compliance with current international laws and policies, as outlined in the Guide for the Care and Use of Laboratory Animals (National Academy Press, 1996). Additionally, these procedures were approved by the Local Ethics Committee for Animal Use at the National Center for Biotechnology.

### 2.2. Cell Culture of Rat Adipose-Derived Pericytes

Pericytes were separated from rat adipose tissue with fluorescence-activated cell sorting (FACS) as previously described [29]. Rat pericytes were expanded in minimum essential medium alpha (Alpha-MEM) (Gibco, Grand Island, NY, USA) supplemented with 10% fetal bovine serum (FBS) (Gibco, Carlsbad, CA, USA) and 50 µg/mL gentamicin at 37 °C with 5% CO_2_. Cell passaging was carried out when a monolayer of adherent cells reached 70–80% confluence.

### 2.3. Preparation of the HCF Hydrogel

The HCF hydrogel was fabricated in accordance with the previously outlined protocol [30]. In summary, the hydrogel was created by combining heparin-conjugated fibrinogen (40 mg/mL), plasminogen-free fibrinogen (60 mg/mL), aprotinin (100 KIU/mL) (Sigma, Burlington, MA, USA), human thrombin (500 IU/mg) (Sigma, Burlington, MA, USA), and calcium chloride (6 mg/mL) (Sigma, Burlington, MA, USA). All components of the hydrogel were dissolved in phosphate-buffered saline (PBS) (Gibco, Carlsbad, CA, USA) and subsequently sterilized using a polyethersulfone membrane filter with a pore size of 0.45 µm (TPP, Trasadingen, Switzerland).

The fibrin hydrogel was created by mixing plasminogen-free fibrinogen (100 mg/mL), aprotinin (100 KIU/mL), human thrombin (500 IU/mg) and calcium chloride (6 mg/mL) in PBS.

### 2.4. Scanning Electron Microscopy (SEM)

The HCF hydrogel sample was dehydrated using a freeze dryer Martin Christ Beta 2–8 LD plus (Martin Christ Gefriertrocknungsanlagen GmbH, Osterode am Harz, Germany). The morphology of the freeze-dried HCF hydrogel was then analyzed using SEM (Auriga Crossbeam 540, Carl Zeiss, Oberkochen, Germany) after being coated with a 10 nm layer of gold.

### 2.5. Measurement of In Vitro BMP-2 Release Kinetics

The release kinetics of BMP-2 from hydrogels were assessed using ELISA Quantification Kits (Abcam, Cambridge, UK), following the manufacturer’s instructions. Fibrin and HCF hydrogels containing BMP-2 (1 µg/mL) were immersed in 1 mL of PBS at 37 °C under orbital shaking at 50 rpm for 28 days. The supernatants were collected every 24 h, and the gels were replenished with fresh PBS. The concentration of released BMP-2 was quantified using a BioRad 680 plate reader (Biorad, Steenvoorde, France) at wavelengths of 450 nm and 540 nm.

### 2.6. Bioactivity of Released BMP-2 In Vitro

Osteoblasts were isolated from the calvaria of newborn Wistar rats (1 day old) by a digestive enzymatic process. Cells were seeded at a density of 5 × 10^4^ cells/well in each well of a six-well culture plate (TPP, Trasadingen, Switzerland). HCF hydrogels loaded with BMP-2 (1 μg) were placed on a Transwell cell culture insert (Corning, Steuben County, NY, USA) in culture dishes. The hydrogels and cells were cultured in alpha-MEM containing 10% FBS (Gibco, Grand Island, NY, USA) and 50 µg/mL gentamicin. ALP activity was evaluated by means of *p*-nitrophenol phosphate (Sigma-Aldrich, Burlington, MA, USA). The cell monolayer was rinsed three times with PBS and lysed in alkaline lysis buffer followed by three freeze–thaw cycles at −20 °C and 37 °C. Cell lysates were incubated in glycine buffer supplemented with *p*-nitrophenol phosphate. In order to stop the reaction, 3 N of NaOH were added to each well after 30 min.

The absorbance of *p*-nitrophenol was measured at 405 nm. The total protein concentration was measured using the Bradford reagent (Sigma-Aldrich, Burlington, MA, USA). ALP activity was then normalized to the total cellular protein.

### 2.7. Mineralization Assay

Rat ADPs were cultured with the prepared culture media for differentiation for 7, 14, and 21 days. In order to identify calcium deposits, each cell monolayer in the well was rinsed with PBS, preserved in 95% alcohol and stained with 1% alizarin red S for 10 min (Sigma-Aldrich, Saint Louis, MO, USA). The remaining color in the wells was removed by washing with distilled water and aspiration. Images were obtained by using an inverted microscope Avio Observer A1 (Carl Zeiss, Oberkochen, Germany) and processed with the ImageJ software (version 1.53t, NIH, Bethesda, MD, USA).

### 2.8. Osteocalcin and Calcium Assay

The quantification of osteocalcin secretion and total calcium content in ADPs was performed with a human osteocalcin ELISA and calcium assay kit (all from Abcam, Cambridge, UK) according to the manufacturer’s instructions.

### 2.9. Rat Calvarial Defect Model and Implantation

Rats were divided into the following five groups (*n* = 5 for each group): (1) control group (without treatment), (2) HCF gel, (3) HCF gel + BMP-2 (1 µg), (4) HCF gel + pericytes (1 × 10^6^ cells) and (5) HCF gel + BMP-2 + pericytes. A critical-sized calvarial defect was performed as previously described by Spicer and colleagues [31]. Briefly, rats were anesthetized with 2% isoflurane using a rodent anesthesia machine R540 (RWD, Shenzhen, Guangdong, China). After treatment with 10% povidone–iodine and additional local anesthesia with the subcutaneous injection of 1% lidocaine in the sagittal part of the skull, a longitudinal 1.5 cm incision down to periosteum was made from the nasal bone to just caudal to the middle sagittal crest. One 8 mm diameter circular transosseous defect was made in the parietal bone using a surgical trephine bur. The drilling site was washed with sterile PBS supplemented with 100 µg/mL gentamicin to chill the bone defect and remove bone chips. The calvaria defects were filled with or without HCF gel according to the group arrangement. Next, skin incisions were sutured and sterilized with 10% povidone–iodine. To mitigate the risk of potential infectious complications, the rats were administered intramuscular injections of 3 mg/kg gentamicin for three days following surgery.

### 2.10. Micro-CT Analysis

The animals were euthanized 12 weeks after the HCF hydrogel implantation, and their rat skulls were harvested and fixed in 4% paraformaldehyde.

Micro-CT imaging (IVIS Spectrum CT, Perkin Elmer, Shelton, CT, USA) was conducted on the calvaria sample to assess bone formation. The imaging parameters included a voltage of 50 kV, a current of 1 mA, and a voxel size of 150 µm. A 440 Al filter was used, with a field of view (FOV) of 12 × 12 × 13 cm and binning set to 4. The total imaging time was 140 s, resulting in an approximate dose of 52.8 mGy per scan. Three-dimensional reconstruction was performed using Living Image 4.3.1 software (Perkin Elmer, Shelton, CT, USA). The 8 mm diameter calvaria defect was selected as the region of interest (ROI) for analysis.

### 2.11. Histological Analysis

All tissue samples from calvarial defects were fixed in 10% neutral-buffered formalin and decalcified with electrolytic decalcification solution for three days (BioVitrum, Saint-Petersburg, Russia). After the dehydration process, the samples were embedded into paraffin and sectioned at 5 μm thickness. The cross-sectioned samples were stained with 1% alizarin red S (Sigma-Aldrich, Burlington, MA, USA). The stained sections were examined with an Axio Scope A1 upright microscope (Carl Zeiss, Oberkochen, Germany) to measure the new bone tissue formation in the defect area. All acquired images were processed using ImageJ software (version 1.53t, NIH, Bethesda, MD, USA).

### 2.12. Statistical Analysis

All obtained data are presented as mean ± SD. The statistical significance was calculated using one-way ANOVA followed by Bonferroni’s multiple comparison tests. A significance level of *p* < 0.05 was considered statistically significant. Quantitative data were expressed as mean ± standard error (SE). Statistical data analysis was performed with software Statistica 6.0 (StatSoft, Tulsa, OK, USA).

## 3. Results

### 3.1. Characterization of HCF Hydrogel

In the present study, the HCF hydrogel was prepared by combining the heparin-conjugated fibrinogen, the human fibrinogen, the human thrombin, the aprotinin, and the calcium chloride. The gelation time of HCF hydrogel was 3 min at room temperature. The gross morphology of the HCF hydrogel after the gelation process is presented in Figure 1A. HCF hydrogel surface morphology was examined by SEM (and Figure 1B). The SEM data showed that the prepared HCF hydrogel has an open interconnected pore morphology and a macroporous structure (Figure 1B). The presence of pores in the HCF hydrogel ensures cell attachment and their further proliferation. Next, we assessed the release kinetics of BMP-2 from the HCF hydrogel using ELISA. The BMP-2 release profile from HCF hydrogels is presented in Figure 1C. ELISA showed a burst release of 45.2 ± 3.8% of total BMP-2 from fibrin hydrogel during 2 days and then 90.4 ± 5.4% of total BMP-2 release after 10 days of incubation in PBS. On the contrary, the release of BMP-2 from the HCF hydrogel was significantly sustained. During the 2 days, 26.5 ± 2.2% of the total BMP-2 from the HCF hydrogel was released, and 66.7 ± 3.5% of BMP-2 was detected on day 10. A complete release of BMP-2 from the HCF hydrogel was observed on day 28. Therefore, our results indicate that the HCF hydrogel demonstrated the controlled release of BMP-2 in a sustained manner.

Next, we examined the bioactivity of released BMP-2 from HCF hydrogel using rat neonatal carvarial osteoblasts. The bioactivity of fresh and eluted BMP-2 was assessed by determining its ability to induce intercellular ALP activity in the osteoblasts at 7 and 14 days. As shown in Figure 1D, fresh and eluted BMP-2 significantly enhanced ALP activity in rat calvarial osteoblasts compared to the control group (without BMP-2). However, there was no difference in ALP activity between the two experimental groups. Thus, these data showed that the HCF + BMP-2 group retained its bioactivity and had a similar effect on the secretion of ALP in cultured calvarial osteoblasts to the BMP-2 group.

### 3.2. BMP-2 Induces Osteogenic Differentiation of ADPs

In this investigation, we examined the effects of eluted BMP-2 on the in vitro osteogenesis of rat ADPs. The osteogenic differentiation of ADPs was examined by ALP activity, osteocalcin expression and calcium deposition. Compared to the control group, the HCF + BMP-2 group demonstrated a significantly higher expression of key osteogenic markers such as osteocalcin and ALP at 7, 14, and 21 days of culture. (Figure 2A,B). The measurement of calcium deposition and alizarin red S staining was performed at 14 and 21 days after induction to evaluate the osteogenesis in ADPs. According to these results, the total calcium content and calcium deposition in the HCF + BMP-2 group increased substantially with differentiation induction compared to the control group (Figure 2C,D).

### 3.3. Effects of HFC Hydrogel Containing BMP-2 and ADPs on Bone Tissue Regeneration

To assess the efficacy of HCF hydrogels containing BMP-2 and/or ADPs, we implanted them into rat calvarial critical-sized defects. The bone formation in calvarial defects was evaluated using micro-CT and histological analysis at three months postsurgery. As shown in Figure 3A, micro-CT images of the control group (without hydrogel) and HCF hydrogel group revealed minimal bone tissue regeneration on the edge of the calvarial defects.

In contrast, the implantation of HCF hydrogel with BMP-2 and HCF hydrogel with encapsulated pericytes into cranial defects resulted in a significant regeneration of the defect with bone tissue formation after 3 months. In the HCF hydrogel + BMP-2 and HCF hydrogel + pericytes groups, defect regeneration with calcified tissue averaged 74% and 68%, respectively (Figure 3D). The quantification of bone volume in these experimental groups using micro-CT revealed significantly higher bone volumes at 12 weeks compared to the control and HCF hydrogel groups (Figure 3C). However, the greatest regenerative effect was found after the co-delivery of pericytes and BMP-2 into the calvarial defect. In the HCF hydrogel + BMP-2 + pericytes group, defect regeneration with calcified tissue averaged 79% and bone volume averaged 92% (Figure 3C,D).

After micro-CT analysis, a histological analysis of rat skull defect samples was carried out after staining with alizarin red S. In tissue sections stained with alizarin red S, the red color indicates mineralized collagen. As shown in Figure 3B, the histological findings were consistent with the X-ray observations. In both the control group and the HCF hydrogel group, there was limited new bone formation, and the areas of calvarial defects were predominantly filled with connective tissue. In the HCF hydrogel + BMP-2 and HCF hydrogel + pericytes groups, the degree of bone tissue formation was noticeably higher compared to the control group. In contrast, the combined delivery of pericytes and BMP-2 in the HCF hydrogel significantly promoted new bone regeneration. The areas of calvarial defects were almost completely covered with calcified bone tissue after three months of healing.

## 4. Discussion

Critical-sized bone defects that occur after surgery or severe mechanical trauma do not undergo spontaneous recovery and necessitate substantial bone grafting and time for restoration. The regeneration of critical-sized bone defects is currently a focal point of research with promising prospects resting on the utilization of tissue engineering materials. These materials aim to restore the structural and functional attributes of compromised bone tissue by leveraging stem cells, growth factors, and natural polymers in combination. The successful use of drugs or bioactive molecules intended to accelerate the regeneration of damaged tissue requires an appropriate carrier that can localize the biomaterial to the target site and protect the encapsulated drug from premature inactivation [6]. In this investigation, we fabricated HCF hydrogels with ADPs and BMP-2, which plays a pivotal role in osteogenesis, bone formation and remodeling. The use of heparin for conjugation with fibrinogen was due to the fact that heparin is a highly sulfated, anionic polysaccharide consisting of repeating glycosaminic and uronic acid residues, which determines its anticoagulant properties. Due to the presence of a significant amount of negatively charged sulfate and carboxyl groups, the heparin molecule is a strong natural polyanion capable of forming complexes with many protein and synthetic compounds of a polycationic nature that carry a total positive charge [32]. In this regard, the sulfate groups of heparin can interact with a number of heparin-binding proteins such as BMPs, forming strong polyanionic complexes [33]. These complexes form a stable form providing protection of heparin-binding proteins from proteolytic degradation, long-term preservation of their biological activity and the delayed release of growth factors from the hydrogel [10,34]. Indeed, our in vitro release assay demonstrated a sustained release of BMP-2 from the HCF hydrogel over a 4-week period. Furthermore, we confirmed the biological activity of the eluted BMP-2, as evidenced by its ability to stimulate ALP activity in rat neonatal osteoblasts.

Having studied the efficiency of eluted BMP-2 factor toward the osteogenic differentiation of rat ADPs, we found that BMP-2 significantly increased the intracellular ALP activity, an early marker of osteogenesis in ADPs at 7, 14 and 21 days, in contrast to the control group. A marked increase in the late markers defining the osteogenic differentiation process, namely, osteocalcin, total calcium and the degree of mineralization in ADPs was revealed within 21 days of culture in the medium with the eluted BMP-2 factor. Thus, the data obtained suggest that the HCF hydrogel, even with a low dose of BMP-2, can drive the osteogenic differentiation of ADPs without the need for additional soluble osteogenic signals present in the induction culture media.

After the conduction of in vitro studies that confirmed the effectiveness of HCF hydrogel on ADPs and osteoblasts, a preclinical test of the therapeutic efficacy of implantation of an injectable HCF hydrogel containing BMP-2 and ADPs was conducted on the rat model with critical-sized calvarial defects. The results of our study showed that both HCF hydrogel with BMP-2 and HCF hydrogel with ADPs are able to significantly increase the regeneration of the defect with calcified tissue, as well as the bone content in the defected area, in contrast to the control group. The greatest effect was found after the co-delivery of ADPs and BMP-2 into the critical-sized calvarial defect. As noted, in the group of HCF hydrogel + BMP-2 + pericytes, defect regeneration with calcified tissue averaged 79% and bone tissue volume averaged 92% after 3 months of hydrogel implantation. The data obtained using micro-CT was confirmed by histological studies of sections stained with alizarin red. These results are consistent with the previous study of James et al. reporting that human adipose-derived perivascular cells loaded onto a PLGA scaffold were able to accelerate the restoration of the critical-sized cranial defects in a mice model [17]. Moreover, it was demonstrated that the combined application of BMP-2 and perivascular cells significantly promoted osteogenesis and bone formation [18,35]. Taken together, these data indicate that the combined application of perivascular cells and BMP-2 may have a synergistic effect, leading to a significant enhancement of bone regeneration.

Thus, the use of HCF hydrogel containing BMP-2 and ADPs may improve patient outcomes, such as enhanced bone regeneration, reduced healing time, and improved functional recovery. However, the importance of safety and biocompatibility assessments in preclinical studies has to be considered to ensure the feasibility and safety of translating the approach into clinical trials. Moreover, the translation of this strategy into clinical practice may be challenging regarding industrial aspects. These may include regulatory approval, the standardization of manufacturing processes, and cost-effectiveness. Compared to existing treatment options, considering factors such as material and manufacturing costs, access to the market might be cost-intensive; however, potential healthcare savings associated with improved patient outcomes may outweigh the aforementioned aspects.

Our obtained HCF hydrogel containing another kind of soluble factor or particles can be used for the biomedical industry. Such HCF hydrogels can function as a transporter role for target drug delivery. Our developed HCF hydrogel that contained the pericytes and BMP-2 or other definite cell types and definite factors is promising for tissue engineering and for use as a bioassay system or biosensor [36,37]. Also, HCF hydrogel can be used as the model for drug testing, allowing avoiding the use of animals [38]. Thus, our developed HCF hydrogel holding the pericytes and BMP-2 as the three-dimensional scaffold possesses great potential for biomedical engineering.

The literature search did not show any work on studying the effect of exactly pericytes within HCF hydrogel and BMP-2. Thus, this work is the first in this field. In our previous review article, we found that pericytes maintaining the properties of tissue restoration and blood pressure modulation can contribute to bone regeneration by paracrine abilities, osteogenic differentiation and vascularization processes [39]. Also, we reported on the clinical use of pericytes, which can be considered for fracture curing and improving osteogenesis. The combined delivery of BMP-2 and pericytes was markedly efficient for bone regeneration [39].

In summary, the application of HCF hydrogel with BMP-2 and ADPs represents a promising approach for bone repair in various clinical scenarios. As with any advanced therapeutic approach, a multidisciplinary approach involving close collaboration between researchers, clinicians, regulatory bodies, and industry partners will be crucial for successful clinical translation in orthopedics and traumatology.

## 5. Conclusions

In conclusion, our in vitro study demonstrated that the HCF hydrogel exhibited the controlled release of BMP-2 in a sustained manner. Furthermore, BMP-2 released from the HCF hydrogel retained its biological activity and effectively induced osteogenesis in rat calvarial osteoblasts and ADPs. The results of the in vivo study showed that the combined application of allogeneic ADPs with BMP-2 enhances the regeneration of critical-sized calvarial defects in rats by promoting significant bone formation and mineralization. Therefore, our findings suggest that the combined delivery of ADPs and BMP-2 in HCF hydrogel has the potential to serve as an alternative biomaterial for the restoration of large bone defects.

## Figures and Tables

**Figure 1 bioengineering-11-00437-f001:**
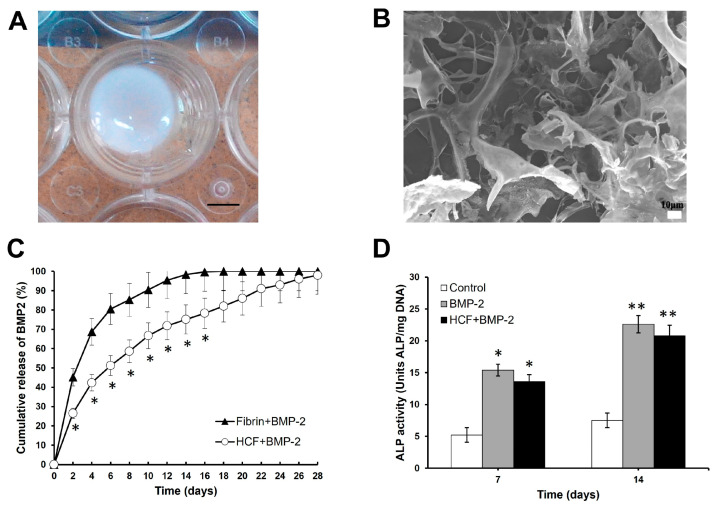
(**A**) Representative image of gross morphology of HCF hydrogel. Scale bar is 5 mm. (**B**) The SEM image of surface morphology of HCF hydrogel. Scale bar is 10 µm. (**C**) Profile of the cumulative release of BMP-2 from fibrin and HCF hydrogels. (**D**) ALP activity in cranial osteoblasts after stimulation by eluted BMP-2. The positive control is the culture medium containing 100 ng/mL of BMP-2. The values are represented as the mean ± SD (*n* = 5). * *p* < 0.05, ** *p* < 0.01.

**Figure 2 bioengineering-11-00437-f002:**
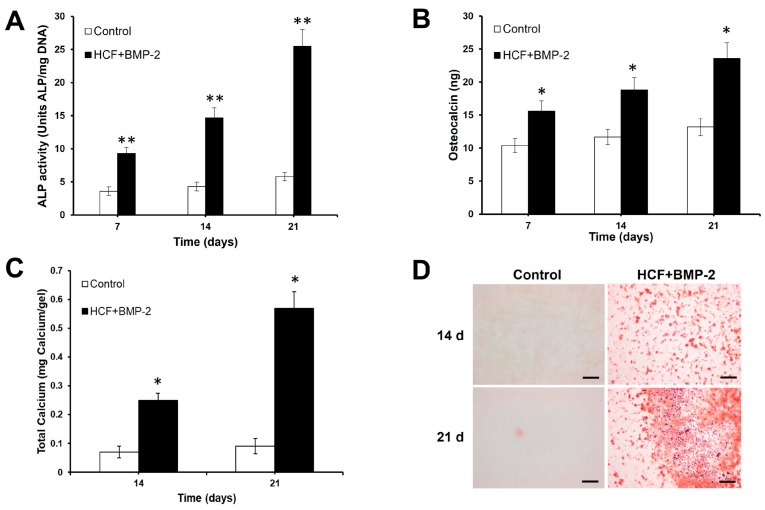
The effects of eluted BMP-2 on osteogenic differentiation of rat ADPs. (**A**) ALP activity in rat ADPs after stimulation by eluted BMP-2. (**B**) Expression of octeocalcin in rat adipose-derived pericytes. (**C**) Calcium deposition by rat ADPs. (**D**) Alizarin red staining of rat ADPs after induction osteogenic differentiation. Scale bar is 100 µm. The values are represented as the mean ± SD (*n* = 5). * *p* < 0.05, ** *p* < 0.01.

**Figure 3 bioengineering-11-00437-f003:**
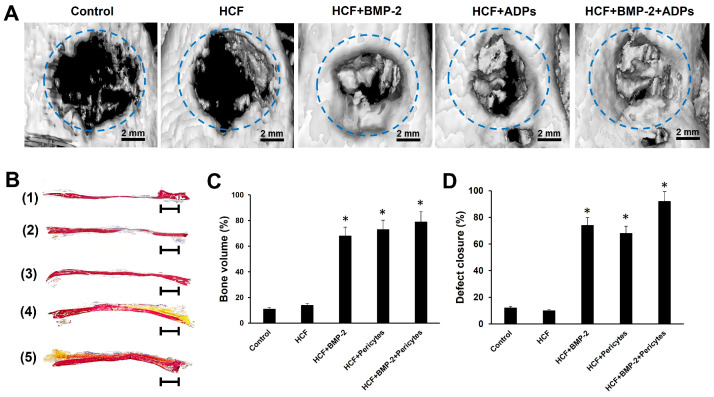
HCF hydrogel containing pericytes and BMP-2 promotes the regeneration of new bone tissue in rat calvarial critical-sized defect. (**A**) Three-dimensional (3D) micro-CT images of rat calvarial critical-sized defects after three months of hydrogel implantation. Scale bar is 2 mm; (**B**) histological sections of the defective area of the skull after staining with alizarin red S: (1) Control, (2) HCF hydrogel, (3) HCF hydrogel + BMP-2, (4) HCF hydrogel + pericytes, (5) HCF hydrogel + BMP-2 + pericytes. Scale bar is 1 mm; (**C**) analysis of the volume of formed bone tissue; (**D**) percentage of bone defect closure. The values are represented as the mean ± SD, * *p* < 0.05.

## Data Availability

All data generated or analyzed during this study are included in this article.

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
