# Peer review of "Repair of Rat Calvarial Critical-Sized Defects Using Heparin-Conjugated Fibrin Hydrogel Containing BMP-2 and Adipose-Derived Pericytes"

_bioengineering, 2024, doi:10.3390/bioengineering11050437_

Round 1
Reviewer 1 Report (Previous Reviewer 1)
Comments and Suggestions for Authors
The paper looks well-structured after revision. It may be considered for publication.
Comments on the Quality of English LanguageEnglish has been improved
Author Response
Dear Reviewer,
Thank you very much for comments and recomendations that improved our research article!
Reviewer 2 Report (Previous Reviewer 2)
Comments and Suggestions for Authors
The manuscript had been well improved compared to the original version. Thanks for your resubmission.
Author Response
Dear Reviewer,
Thank you very much for comments and recomendations that improved our research article!
This manuscript is a resubmission of an earlier submission. The following is a list of the peer review reports and author responses from that submission.
Round 1
Reviewer 1 Report
Comments and Suggestions for Authors
In this paper, the authors prepared heparin-fibrin conjugated hydrogels and encapsulated BMP-2 and adipose-derived pericytes for the repair of critical-sized calvarial defects. Both in vitro and in vivo analyses were conducted. The study is well-designed, so I recommend the publication of this paper in Bioengineering, after a few concerns are addressed:
1. Introduction: The authors stated that “Both physical and chemical synthesis methods have been applied to create injectable hydrogels.” More references need to be added to support this statement. For example, you can reference “Carbohydrate polymers, 2017, 156: 403-408” for physical synthesis; and “ACS Appl. Mater. Interfaces 2020, 12, 28, 31079–31089” for chemical synthesis.
2. The authors stated that “Despite the fact that injectable hydrogels obtained by different methods have been studied for decades, there are currently no appropriate injectable hydrogels that would be effectively used in clinical regenerative medicine.” Is this statement accurate? Several injectable hydrogels have been approved by the FDA and applied in clinical settings. Please clarify or revise this statement accordingly.
3. I agree with the authors that the choice of the optimal cell source, growth factors, and carrier is important for this application. The authors listed their reasons for choosing the cell source and growth factors. However, could you provide a more detailed explanation for choosing heparin-fibrin hydrogels in this study? Considering that fibrin might be relatively expensive for clinical translation, it would be helpful to address the cost-effectiveness of this choice.
4. More descriptive statements regarding Figure 1A need to be included in the manuscript to provide better context and understanding of the figure.
5. For your ELISA results, please present the data as mean ± SD to ensure consistency and clarity.
6. Maintain consistency throughout the manuscript regarding figure references. Sometimes "Figure 1B" is mentioned, while other times it is "Fig. 1C." Please thoroughly review and ensure consistency in figure labeling.
7. In Figure 3B, please include scale bars to provide a reference for the size or dimensions of the displayed objects.
Comments on the Quality of English LanguageEnglish looks good.
Author Response
Responses for referee’s comments
In this paper, the authors prepared heparin-fibrin conjugated hydrogels and encapsulated BMP-2 and adipose-derived pericytes for the repair of critical-sized calvarial defects. Both in vitro and in vivo analyses were conducted. The study is well-designed, so I recommend the publication of this paper in Bioengineering, after a few concerns are addressed:
- Introduction: The authors stated that “Both physical and chemical synthesis methods have been applied to create injectable hydrogels.” More references need to be added to support this statement. For example, you can reference “Carbohydrate polymers, 2017, 156: 403-408” for physical synthesis; and “ACS Appl. Mater. Interfaces 2020, 12, 28, 31079–31089” for chemical synthesis.
- Thank you for comment! We added 2 relevant references as you recommended.
- The authors stated that “Despite the fact that injectable hydrogels obtained by different methods have been studied for decades, there are currently no appropriate injectable hydrogels that would be effectively used in clinical regenerative medicine.” Is this statement accurate? Several injectable hydrogels have been approved by the FDA and applied in clinical settings. Please clarify or revise this statement accordingly.
- Thank you for comment! We revised this statement in the text.
- I agree with the authors that the choice of the optimal cell source, growth factors, and carrier is important for this application. The authors listed their reasons for choosing the cell source and growth factors. However, could you provide a more detailed explanation for choosing heparin-fibrin hydrogels in this study? Considering that fibrin might be relatively expensive for clinical translation, it would be helpful to address the cost-effectiveness of this choice.
- Thanks for your comment! We added detailed information on choosing the heparin-conjugated fibrin hydrogel for our study. Please see our explanation in the Introduction.
- More descriptive statements regarding Figure 1A need to be included in the manuscript to provide better context and understanding of the figure.
- Thank you for comment! We inserted additional information and rewrited description of the results on characterization of HCF hydrogel.
- For your ELISA results, please present the data as mean ± SD to ensure consistency and clarity.
- Thank you for comment! We corrected and presented ELISA data as mean ± SD in the text.
- Maintain consistency throughout the manuscript regarding figure references. Sometimes "Figure 1B" is mentioned, while other times it is "Fig. 1C." Please thoroughly review and ensure consistency in figure labeling.
- Thank you for comment! We corrected this mistake in the manuscript.
- In Figure 3B, please include scale bars to provide a reference for the size or dimensions of the displayed objects.
- Thank you for comment! We inserted scale bars in Figure 3B.
Reviewer 2 Report
Comments and Suggestions for Authors
This paper about the polymeric hydrogel was interesting. The topic was to some degree significant. The paper fell within the scope of Bioengineering, and could be considered for publication after a Major Revision. Please refer to my detailed comments:
1). A scheme about the whole picture of this work could be added at the end of the Introduction Section.
2). The weight of Wistar Rats should be mentioned in Section 2.1.
3). According to Figure 1B, the drug release was quite slow, which consumed up to 3 weeks. Was the drug loading and the released drug content sufficient for disease therapy? 4). Following 3), it would be advisable to offer the drug loading and encapsulation efficiency data.
5). SEM, TEM or AFM should be conducted to explore the microscopic morphology of the hydrogel.
6). Statistical analysis should be performed in Figure 3C and 3D, especially for the last three groups.
7). Please discuss industrial and clinical translation aspects in Section 4.
8). An individual Conclusion Section was missing.
9). The format of References should be unified.
Author Response
Responses for referee’s comments
This paper about the polymeric hydrogel was interesting. The topic was to some degree significant. The paper fell within the scope of Bioengineering, and could be considered for publication after a Major Revision. Please refer to my detailed comments:
1). A scheme about the whole picture of this work could be added at the end of the Introduction Section.
-Thank you for comment! We prepared a schematic illustration of our work. We think that it will be better to use it as graphical abstract in our article.
2). The weight of Wistar Rats should be mentioned in Section 2.1.
- Thank you for comment! We include the information about the weight in Section 2.1.
3). According to Figure 1B, the drug release was quite slow, which consumed up to 3 weeks. Was the drug loading and the released drug content sufficient for disease therapy?
-Thank you for comment! Fibrin and HCF hydrogels were used to study the release of BMP-2. After 48 hours, the release of BMP-2 from fibrin hydrogel was 45.2%, and from HCF hydrogel 26.5%. Almost complete release of growth factors from the fibrin hydrogel was observed on the 14 day of incubation, while from HCF hydrogel complete release was observed on the 28 day. The concentration of BMP-2 was 1 µg. At this concentration, bone tissue restoration was 74%, which is also a good result. However, in the HCF hydrogel+BMP-2+pericytes group, defect regeneration with calcified tissue averaged 79% and bone volume averaged 92%.
4). Following 3), it would be advisable to offer the drug loading and encapsulation efficiency data.
-Thank you for comment! Our objective was to determine the duration of BMP-2 release and investigate its effect on the efficiency of bone tissue repair. We agree with the reviewer's opinion that it would be advisable to offer the drug loading and encapsulation efficiency data. However we did not conduct this study, but perhaps we will conduct it in our future studies.
5). SEM, TEM or AFM should be conducted to explore the microscopic morphology of the hydrogel.
-Thank you for your comment! We added SEM morphology of the HCF hydrogel in Fig. 1.
6). Statistical analysis should be performed in Figure 3C and 3D, especially for the last three groups.
- Thank you for comment! We performed statistical analysis in Figure 3C and 3D. Significant difference from control group was *p < 0.05.
7). Please discuss industrial and clinical translation aspects in Section 4.
Thanks for your comment! We added the discussion on industrial and clinical translation.
8). An individual Conclusion Section was missing.
- Thank you for comment! We added conclusion in the manuscript.
9). The format of References should be unified.
- Thank you for comment! We corrected the references, according to the requirements of the journal.
